# Prevalence of Opioid Use Disorder among Patients with Cancer-Related Pain: A Systematic Review

**DOI:** 10.3390/jcm11061594

**Published:** 2022-03-14

**Authors:** Céline Preux, Marion Bertin, Andréa Tarot, Nicolas Authier, Nathalie Pinol, David Brugnon, Bruno Pereira, Virginie Guastella

**Affiliations:** 1Palliative Care Center, CHU Clermont-Ferrand, F-63000 Clermont-Ferrand, France; cpreux@chu-clermontferrand.fr (C.P.); marion.bertin@uca.fr (M.B.); atarot@chu-clermontferrand.fr (A.T.); dbrugnon@chu-clermontferrand.fr (D.B.); 2Neuro-Dol, Service Pharmacologie Médicale, Centres d’Addictovigilance et Pharmacovigilance, Centre Evaluation et Traitement de la Douleur, Inserm, CHU Clermont-Ferrand, Université de Clermont Auvergne, F-63001 Clermont-Ferrand, France; nauthier@chu-clermontferrand.fr; 3Institut Analgésia, Faculté de Médecine, BP38, F-63001 Clermont-Ferrand, France; 4Observatoire Français des Médicaments Antalgiques/French Monitoring Centre for Analgesic Drugs, Université Clermont-Ferrand, F-63001 Clermont-Ferrand, France; 5Centre de Documentation et Recherche de la Faculté de Médecine, CHU Clermont-Ferrand, Université de Clermont Auvergne, F-63000 Clermont-Ferrand, France; nathalie.pinol@uca.fr; 6Unité de Biostatistiques, Direction de la Recherche Clinique et de l’Innovation, CHU Clermont-Ferrand, F-63000 Clermont-Ferrand, France; bpereira@chu-clermontferrand.fr

**Keywords:** meta-analysis, systematic review, prevalence, opioid use disorder, cancer, chronic pain

## Abstract

Background: The opioid use disorder is an international public health problem. Over the past 20 years it has been the subject of numerous publications concerning patients treated for chronic pain other than cancer-related. Patients with cancer-related pain are also at risk of opioid use disorder. The primary objective of this literature review was to determine the prevalence of opioid use disorder in patients with cancer-related chronic pain. Its secondary objective was to identify the characteristics of these opioid users. Methods: This is a literature review of studies published over the last twenty years, from 1 January 2000 to 31 December 2020 identified by searching the three main medical databases: Pubmed, Cochrane, and Embase. A meta-analysis took account of between and within-study variability with the use of random-effects models estimated by the DerSimonian and Laird method. Results: The prevalence of opioid use disorder was 8% (1–20%) and of the risk of use disorder was 23.5% (19.5–27.8%) with *I*^2^ values of 97.8% and 88.7%, respectively. Conclusions: Further studies are now needed on the prevalence of opioid use disorder in patients treated for cancer-related chronic pain. A screening scale adapted to this patient population is urgently needed.

## 1. Introduction

In oncology, pain is a major discomfort symptom, with a significant impact on quality of life [1,2]. The World Health Organization (WHO) has identified pain as a major issue in the management of cancer patients. The use of opioid analgesics is recommended for the treatment of moderate to severe cancer pain [3].

The prescription of opioid analgesics has increased dramatically in many countries in recent years, as have deaths arising from their use [4]. In France, between 2000 and 2015, these deaths increased by 146%, from 76 to 204 (four deaths per week) [5]. The opioid use disorder is now an international public health problem. Over the past 20 years, it has been the subject of numerous publications concerning patients treated for pain other than cancer-related [6,7,8,9,10]. Various scales have also been developed to help prescribers detect risk of opioid use disorder [11] in the general population.

Patients with cancer-related pain are also at risk of aberrant opioid analgesic behaviors, misuse or addiction [12,13]. Few studies seem to have addressed this risk. There are no screening scales for the risk of opioid use disorder in this population of patients with cancer-related pain.

The primary objective of this literature review was to determine the prevalence of opioid use disorder in patients with cancer-related chronic pain. Its secondary objective was to identify the characteristics of such users.

## 2. Materials and Methods

This study followed the PRISMA guidelines for reporting systematic reviews and meta-analysis (Appendix A). This is a literature review of studies published over the last twenty years, from 1 January 2000 to 31 December 2020. It has been registered on Prospero (CRD42021241784).

### 2.1. Search Strategy

We searched three major medical databases: Pubmed, Cochrane, and Embase. We used the following keywords: chronic cancer-related pain (and synonymous terms: cancer-associated pain, tumor-associated pain, oncology pain), inappropriate drug use, abuse, overdose, addiction, non-medical use, opioid analgesics (and equivalents: full opioid agonists, partial opioid agonists, opioid agonist-antagonists). We specified three search equations (Appendix A) using the keywords, excluding non-cancer pain and non-opioid analgesics. We extracted articles dealing with the following: non-cancer pain, acute pain with no mention of chronic pain, under-prescription of strong opioid analgesics owing to prescribers’ fear of possible misuse, general practitioners’ opinions on opioid prescribing, patients’ opinions on opioid use, articles limited to urine tests to detect misuse, literature reviews, and documentary articles.

The search was conducted by two researchers (CP and MB) on Pubmed, Cochrane et Embase. When there was disagreement, a third reviewer was consulted to reach consensus. (VG). The articles selected by the search engine were first sorted (Phase 1) on title and abstract by two independent readers for each article. A third reader was consulted when there was disagreement between the first two readers. The selected articles were then further sorted (Phase 2) after reading them in full.

### 2.2. Selection Criteria

We drew up a data collection table to record the information we needed to pursue our objectives: prevalence of observed of opioid use disorder and prevalence of risk of opioid use disorder, population characteristics: age, sex, marital status, social precariousness, type of cancer, stage of cancer, pain intensity, psychiatric history, history of addiction, history of sexual abuse, other prescribed treatments: benzodiazepines, other analgesics, type of use disorder, clinical impact, pain intensity, inpatient or outpatient management, prescribed substance, and route of administration. The prevalence of risk of opioid use disorder, not initially searched for, was finally recorded because it occurred frequently in the selected articles.

Each article included in the analysis was read by two independent readers (CP and MB) who filled in data collection tables independently. The two tables were then compared to generate a final table with the data for the analysis. A third reader (VG) was consulted when there was disagreement between the two tables.

### 2.3. Assessment of Methodological Quality 

We assessed the risk of bias for each study using the ‘Methodology checklist 3: cohort studies’ of the Scottish Intercollegiate Guidelines Network [14] together with a tool for assessing risk of bias in prevalence studies designed and evaluated by Hoy D. et al. [15] For each article, two readers (CP and MB) made this assessment independently and then compared their assessments. When there was disagreement between the two reviewers there was a discussion to resolve it, either between them or with a third reviewer (VG).

### 2.4. Statistical Analysis

Statistical analysis was performed with Stata software (version 15 StataCorp, College Station, TX, USA). The prevalence of opioid use disorder (i.e., effect-size noted ES) was estimated using the metaprop command of Stata software. This routine provides procedures for pooling proportions in a meta-analysis of multiple studies and displays the results in a forest plot. The confidence intervals (95% CI) are based on score (Wilson) or exact binomial (Clopper-Pearson) procedures. Our meta-analysis took account of between- and within-study variability. To address the non-independence of data due to study effect, random-effects models were preferred to the usual statistical tests to assess the prevalence of opioid use disorder, using the method of DerSimonian and Laird [16] with the estimate of heterogeneity being taken from the inverse-variance fixed-effect model. A test of whether the summary effect measure was equal to zero is given, together with a test for heterogeneity, i.e., whether the true effect in all studies was the same. For stratified analyses by screening scale (CAGE, ORT, and SOAPP) and by sex, the analogous statistical approach was applied. The results are expressed as prevalence with 95% confidence intervals. Data were not sufficiently robust to conduct meta-regression analysis.

Heterogeneity in the study results was also assessed by forest plots and the *I*^2^ statistic, the most common metric for measuring the magnitude of between-study heterogeneity, and which is easily interpretable. *I*^2^ values range between 0% and 100% and are typically considered low for 25%, moderate for 25–50%, and high for 50% [17]. Publication bias was assessed by funnel plots and confidence intervals, and Egger’s test.

## 3. Results

A flowchart of the search strategy and study selection process are given in Figure 1. We initially identified 1195 studies. After removing duplicates and studies on the first screening, 58 articles were read in full. Finally, a total of 15 studies were included in the present meta-analysis.

The results are organized as follows: Table 1 shows studies characteristics. Although we initially planned to investigate only the prevalence of opioid use disorder, we added the results of the risk of prevalence of opioid use disorder as this was frequent found in the literature.

The prevalence of opioid analgesics use disorder was 8% (1–20%) (Figure 2) and that of risk of use disorder was 23.5% (19.5–27.8%) (Figure 3). In both cases, the heterogeneity of the studies was very high, with *I*^2^ values of 97.8% and 88.7%, respectively.

For the screening of patients at risk of inappropriate opioid analgesic use, the screening scales used in the studies analyzed were the CAGE, ORT, SOAPP-SF (short form), SOAPP-R (revised), and SOAPP-14.

Depending on the screening scale used, the average prevalence ranged between 16.2% and 34.1% (Figure 4): CAGE 19.4% (15.8–23.5%), ORT 16.2% (5.9–30.0%), SOAPP-SF 28.5% (24.7–32.6%), SOAPP-R 34.1% (28.6–39.8%), and SOAPP-14 22.9% (18.2–28.0%).

We found a prevalence of use disorder in males of 17.1% (12.7–21.9%) and in females of 15.5% (11.3–20.3%) (Figure 5). In comparison, the prevalence of risk of use disorder was 24.7% (21.1–28.5%) in males and 17.5% (13.2–22.4%) in females (Figure 6). Here again, the heterogeneity of the studies was very high, with *I*^2^ = 85.5% for females and *I*^2^ = 66.2% for males.

Given the heterogeneity of the studies and lack of data, we could not define a profile of patients at greater risk of opioid analgesics use disorder.

## 4. Discussion

This systematic review and meta-analysis identified a total of 15 studies reporting rates of opioid use disorders among patients with cancer-related pain. Ours is the first literature review on the strong opioid use disorder to manage chronic cancer-related pain. We selected our articles from the three main medical databases (PubMed, Cochrane and Embase). Our inclusion and non-inclusion criteria were sufficiently precise to obtain a satisfactory percentage agreement between the reviewers (CP, MB and AG) (90%). Our analysis found the prevalence of opioid use disorder in patients with cancer-related chronic pain to be 8% (1–20%). However, the very high statistical heterogeneity *I*^2^ = 97.8% rules out any conclusive interpretation. Although prevalence of the risk of use disorder was not initially searched for, it was found as a data item in the studies. It was 23.5% (19.5–27.8%) (*I*^2^ = 88.7%).

Certain limitations hinder the interpretation of our results. First, we chose a broad definition of opioid use disorder that included all the terms used and so increased the number of studies selected at the expense of their homogeneity. We found several terms in the literature that could meet the definition of use disorder [11]. The definition used varied according to the article. Some authors consider only abuse [18], misuse [31] or ‘chemical coping’ [22]. Others take a broader definition integrating misuse, aberrant behaviors linked to strong opioid analgesics [20] and even addiction and dependence [21]. How data were collected varied among studies. In retrospective studies, records were used (Table 1), often leading to a marked bias because records were incomplete. In prospective studies (Table 1), consultations and self-questionnaires were used, which could also lead to information bias, as patients tend to minimize their use disorders. The populations included in each study differed in age, sex, location, and type of cancer. The study of Garcia et al. [29] concerns only females with gynecological cancers. That of Ehrentraut et al. [20] concerns adolescents with an average age of 16.3 years. The other studies concerned only adults. Location varied: Italy for the study of Mercadante et al., [19] Spain for that of Núñez Olarte et al., [21] Egypt for that of Bashandy et al. [32] and the USA for the others.

The primary outcome measure, for prevalence of risk of misuse, differed between studies, depending on the screening scale used. Figure 4 suggests a difference in sensitivity between screening scales. A trend was seen in the characteristics of opioid disorder users: hematological and head and neck cancers seemed more often represented. However, the very high heterogeneity of the studies here again prevents any conclusive interpretation.

Second, there is a paucity of data for analysis, in part because we did not include articles in which the type of pain was not specified, and studies based on urine tests for toxicants. These qualitative tests are counted positive if they detect the presence of other substances such as alcohol or benzodiazepines. These tests are carried out at the request of physicians, especially when there is doubt about of opioid use disorder. We concluded that there was too great a risk of overestimating the prevalence of opioid use disorder. We hoped to identify the characteristics of opioid disorder users, but again lacked data. Certain characteristics (such as a history of addiction, psychiatric disorders, and sexual abuse) were absent in the studies that found opioid use disorder. They were present in the studies that assessed the risk of opioid analgesics use disorder, because they were obtained from the screening questionnaires. However, these led to a major information bias, making them unusable. More objective data such as marital status, type of substance used, and duration of opioid treatment, were too incomplete for any relevant statistical calculation.

Third, our study does not tell us how much opioid use disorder was due to prescription error or insufficient therapeutic education, and how much could be ascribed to a patient profile more prone to such behavior.

Fourth, one of the areas of weakness of our study is to be a descriptive analysis that does not compare anything neither use nor use disorder. The very low number of studies does not permit that.

In the literature review by Hojsted J et al. [33], the prevalence of opioid addiction in patients treated for chronic pain is estimated at 0–7.7% in the cancer population. This is a slightly lower prevalence than ours, with a definition that took into account only one type of inappropriate use.

Regarding our secondary endpoints, in the study by Jairam et al. [34] opioid overdoses were more frequent in patients with certain cancers such as head and neck cancer and myeloma. These are also the two cancer families for which we found a higher prevalence, but our analysis did not enable us to compare these figures with those of other cancers. In the study by Madadi et al. [35], males were over-represented in deaths related to opioid analgesics. This sex difference was not found in our study. However, the populations were different (cancer population vs. general population). In addition, we were studying opioid use disorder, which is not limited to overdose.

## 5. Conclusions

In summary, our meta-analysis shows that the prevalence of opioid use disorder among patients with cancer-related pain is still largely unknown. Although we found a prevalence of 8% (1–20%) use disorder, the heterogeneity of the studies included in the statistical analysis was very high. Further studies are thus needed on the opioid analgesics use disorder in patients treated for cancer-related chronic pain, which is a major public health issue. They should enable the development of a screening tool validated in a cancer population that will help us approach the real prevalence of use disorder.

## Figures and Tables

**Figure 1 jcm-11-01594-f001:**
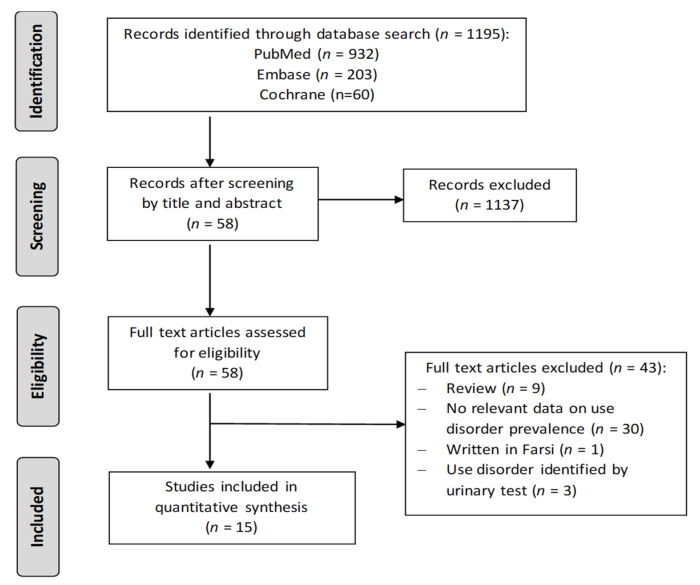
Flowchart of the study selection.

**Figure 2 jcm-11-01594-f002:**
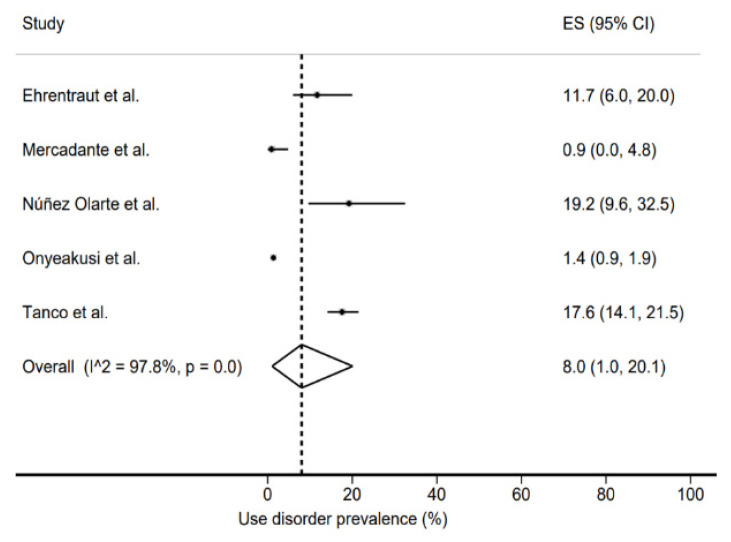
Use disorder prevalence [18,19,20,21,22].

**Figure 3 jcm-11-01594-f003:**
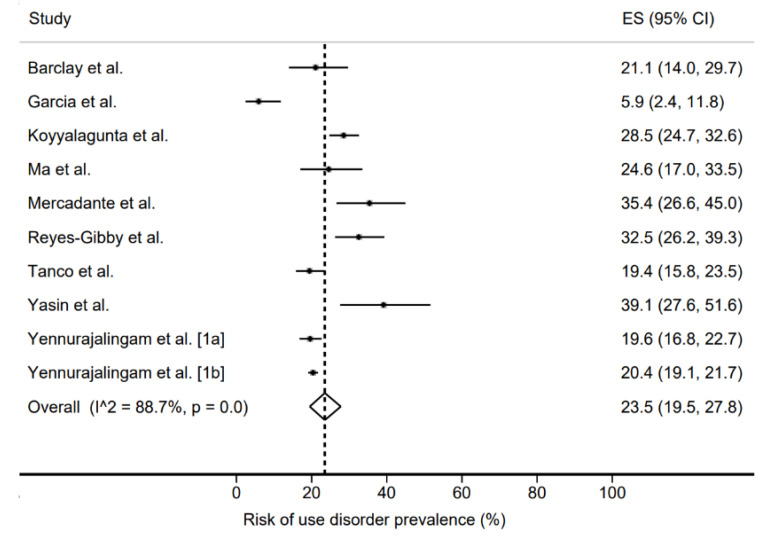
Risk of use disorder prevalence [19,22,23,24,25,26,27,28,29,30].

**Figure 4 jcm-11-01594-f004:**
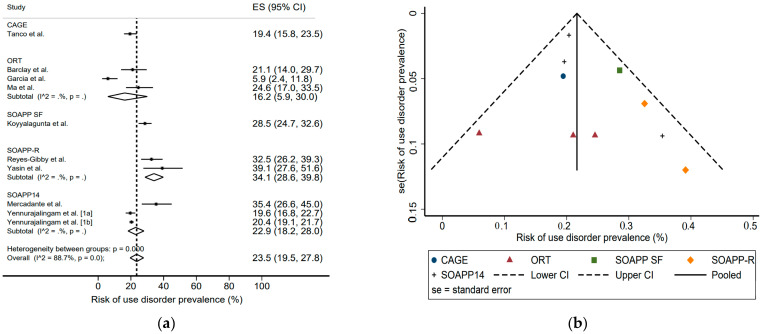
Risk of use disorder prevalence according to screening scale: (**a**) forest plot, (**b**) funnel plot [18,19,22,23,25,26,27,28,29,30].

**Figure 5 jcm-11-01594-f005:**
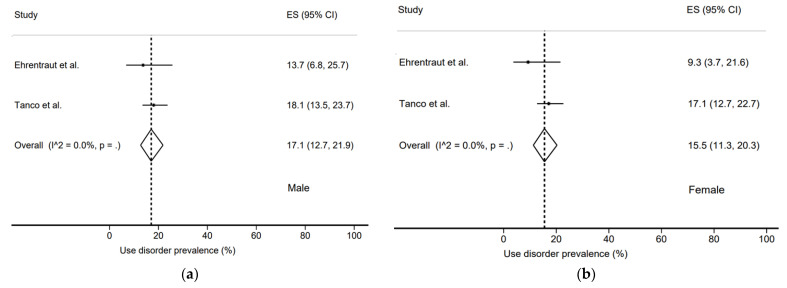
Use disorder prevalence by sex: (**a**) Male; (**b**) Female [20,22].

**Figure 6 jcm-11-01594-f006:**
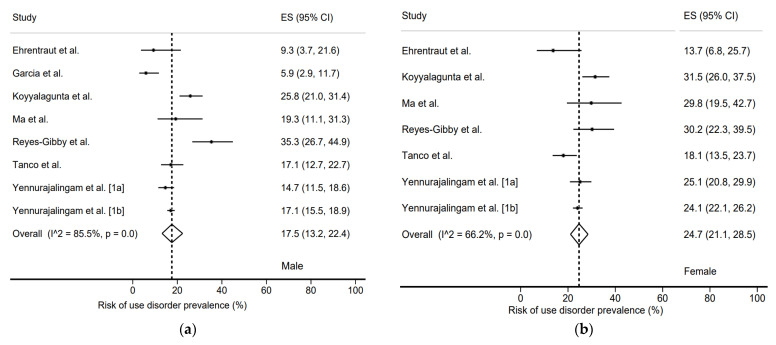
Risk of use disorder prevalence by sex. (**a**) Male [18,20,22,23,25,26,28,29]; (**b**) Female [18,20,22,23,25,26,28].

**Table 1 jcm-11-01594-t001:** Characteristics of studies included in the meta-analysis.

Authors	Country	Type	ART	Total	Use Disorder Prevalence		Scale	Total Population of Study			Population with Use Disorder		
(Publication Year)					Actual (%)	Risk (%)		Females	Mean Age (SD)	hem/dig/gyn/pn/ mel/hn/genit/br	Females	Mean Age (SD)	hem/dig/gyn/pn/ mel/hn/genit/br
Yennurajalingam et al. [23] (2018)	Texas	Rétro. Mono.	7	729	nr	143 (19.6%)	SOAPP14	382	61 (53–69)	nr/nr/nr/nr/ nr/nr/nr/nr	56	61 (54–67)	nr/nr/nr/nr/ nr/nr/nr/nr
Onyeakusi et al. [18] (2019)	Ohio	Retro. Multic.	6	2665	36 (1.35%)	nr		1560	59.5 (0.26)	nr/nr/nr/nr/ nr/nr/nr/nr			nr/nr/nr/nr/ nr/nr/nr/nr
Yennurajalingam et al. [24] (2020)	Texas	Rétro. Mono.	8	3588	nr	731 (20.4%)	SOAPP14	1906	62 (52–70)	178/646/256/573/ 197/572/417/485	326	61 (51–68)	30/130/43/132/ 48/130/96/68
Mercadante et al. [19] (2020)	Italy	Prosp. Multic.	6	113	0	40 (35.4%)	SOAPP14	54	68 (13)	nr/31/nr/26/ nr/nr/26/13			nr/nr/nr/nr/ nr/nr/nr/nr
Passik et al. [31] (2006)	Indiana	Prosp. Multic.	6	100	nr	nr		62	60.1 (13.87)	nr/nr/nr/nr/ nr/nr/nr/nr			nr/nr/nr/nr/ nr/nr/nr/nr
Koyyalagunta et al. [25] (2013)	Texas	Prosp. Mono.	6	522	nr	149 (28.5%)	SOAPP SF	271	54	78/94/nr/57/ nr/78/nr/57	70	50 (±14)	nr/nr/nr/nr/ nr/nr/nr/nr
Ma et al. [26] (2014)	Californie	Rétro. Mono.	6	114	nr	28 (24.6%)	ORT	57	53 (14.3)	21/33/11/4/ nr/8/7/13	11	52.5 (8.3)	4/6/4/3/ nr/2/2/3
Barclay et al. [27] (2014)	Virginie	Rétro. Mono.	5	114	nr	24 (21%)	ORT	73	53	12/13/18/14/ nr/14/2/20			nr/nr/nr/nr/ nr/nr/nr/nr
Ehrentraut et al. [20] (2014)	Tennessee	Rétro. Mono.	6	94	11 (11.7%)	nr	AOB list	43	16.3 (0.28)	47/nr/nr/nr/ nr/nr/nr/nr	4	17.4 (2.8)	7/nr/nr/nr/ nr/nr/nr/nr
Reyes-Gibby et al. [28] (2016)	Texas	Prosp. Mono.	4	209	nr	68 (32.5%)	SOAPP-R	102	54.2 (13.07)	53/15/nr/18/ nr/nr/nr/24	36	52.1	19/2/nr/4/ nr/nr/nr/9
Garcia et al. [29] (2017)	Virginie	Prosp. Mono.	7	118	nr	7 (6%)	ORT	118	57	nr/nr/118/nr/ nr/nr/nr/nr	7	47 (42–56)	nr/nr/7/nr/ nr/nr/nr/nr
Yasin et al. [30] (2019)	Pennsylvanie	Rétro. Mono.	5	69	nr	27 (39.1%)	SOAPP-R	41	nr	nr/nr/nr/nr/ nr/nr/nr/nr			nr/nr/nr/nr/ nr/nr/nr/nr
Núñez Olarte et al. [21] (2018)	Espagne	Rétro. Mono.	7	52	10 (19.2%)	nr		nr	nr	nr/nr/nr/nr/ nr/nr/nr/nr	5	58,1	nr/4/nr/2/ nr/1/1/1
Bashandy et al. [32] (2016)	Egypte	Prosp. Mono.	Abstr	nr	nr	76%	COMM	nr	nr	nr/nr/nr/nr/ nr/nr/nr/nr			nr/nr/nr/nr/ nr/nr/nr/nr
Tanco et al. [22] (2015)	Texas	Rétro. Mono.	7	432	76 (18%)	84 (19.4%)	CAGE	216	57 (49–65)	26/70/nr/86/ nr/48/nr/60	37	53 (46–60)	8/8/nr/15/ nr/14/nr/10

Note. nr = not reported; Type of study: Retro = retrospective; Mono = monocentric; Multic = multicentric; ART = Assessing Risk Tool [15]; Type of cancer: hem = hematologic, dig = digestive, gyn = gynecologic, pn = pneumologic, mel = melanoma, hn = head and neck, genit = genital, br = breast.

## Data Availability

Not applicable.

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
