# Peer review of "Prevalence of Opioid Use Disorder among Patients with Cancer-Related Pain: A Systematic Review"

_jcm, 2022, doi:10.3390/jcm11061594_

Round 1

Reviewer 1 Report

Metaanalysis about opiod use in cancer patients. One limitation you may cite is the diference between opioid disorder and use. Well designed and explained.

One of the areas of weakness is to be a descriptive analysis that not compare anything (use vs disorder) (opioid treatment vs coadyuvants antinociceptive agents)...

Author Response

Point 1 : Metaanalysis about opiod use in cancer patients. One limitation you may cite is the diference between opioid disorder and use

Reply : Indeed, these two notions are different and this work focuses specifically on the notion of “use disorders” as indicated in the title. Further studies are now needed to go on and specifically a screening scale to diagnose the use disorder, adapted to this patient population.

Point 2 : Well designed and explained.

Reply : Thank you for your comment

Point 3 : One of the areas of weakness is to be a descriptive analysis that not compare anything (use vs disorder) (opioid treatment vs coadyuvants antinociceptive agents)...

Reply : yes it is a limit of the study however it was not the objective. Moreover the very low number of studies doesn’t allow us to do that.

Reviewer 2 Report

The auhtors present a well designed study investigating an neglected area of research: the prevalence of and risk for opioid use disorders in patients with cancer pain. This topic is of high clinical relevance, and was only so far evaluated in a few trials. This is the only limitation of this meta analysis, that the number of included studies is small (and the heterogeneity is high). However, the authors addressed this issue in their discussion and they avoid over-interpreting their results. Additionally, the lack of well-designed studies is per se a relevant finding.

The search strategy is concise, but adequate. The study selection and analysis is in accordance with high standards. The discussion is balanced.

The authors should be congratulated for their inspiring work.

Author Response

we thank you for your very nice comments ;  we hope to go on and to study on this  subject in order to better diagnose this use disorder
